# RL Unplugged: A Suite of Benchmarks for Offline Reinforcement Learning

**Caglar Gulcehre**[D,*]    **Ziyu Wang**[G,*]    **Alexander Novikov**[D,*]    **Tom Le Paine**[D,*]

**Sergio Gómez Colmenarejo**[D]    **Konrad Żołna**[D]    **Rishabh Agarwal**[G]    **Josh Merel**[D]

**Daniel Mankowitz**[D]    **Cosmin Paduraru**[D]    **Gabriel Dulac-Arnold**[G]    **Jerry Li**[D]

**Mohammad Norouzi**[G]    **Matt Hoffman**[D]    **Nicolas Heess**[D]    **Nando de Freitas**[D]

**D:** DeepMind      **G:** Google Brain

## Abstract

Offline methods for reinforcement learning have a potential to help bridge the gap between reinforcement learning research and real-world applications. They make it possible to learn policies from offline datasets, thus overcoming concerns associated with online data collection in the real-world, including cost, safety, or ethical concerns. In this paper, we propose a benchmark called RL Unplugged to evaluate and compare offline RL methods. RL Unplugged includes data from a diverse range of domains including games (*e.g.,* Atari benchmark) and simulated motor control problems (*e.g.,* DM Control Suite). The datasets include domains that are partially or fully observable, use continuous or discrete actions, and have stochastic vs. deterministic dynamics. We propose detailed evaluation protocols for each domain in RL Unplugged and provide an extensive analysis of supervised learning and offline RL methods using these protocols. We will release data for all our tasks and open-source all algorithms presented in this paper. We hope that our suite of benchmarks will increase the reproducibility of experiments and make it possible to study challenging tasks with a limited computational budget, thus making RL research both more systematic and more accessible across the community. Moving forward, we view RL Unplugged as a living benchmark suite that will evolve and grow with datasets contributed by the research community and ourselves. Our project page is available on github.

## 1   Introduction

Reinforcement Learning (RL) has seen important breakthroughs, including learning directly from raw sensory streams [Mnih et al., 2015], solving long-horizon reasoning problems such as Go [Silver et al., 2016], StarCraft II [Vinyals et al., 2019], DOTA [Berner et al., 2019], and learning motor control for high-dimensional simulated robots [Heess et al., 2017, Akkaya et al., 2019]. However, many of these successes rely heavily on repeated online interactions of an agent with an environment. Despite its success in simulation, the uptake of RL for real-world applications has been limited. Power plants, robots, healthcare systems, or self-driving cars are expensive to run and inappropriate controls can have dangerous consequences. They are not easily compatible with the crucial idea of

| Task domain | DM Control Suite / Real World RL Suite | DM Locomotion Humanoid | DM Locomotion Rodent | Atari 2600 |
|---|---|---|---|---|

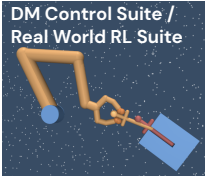   

| | | | | |
|---|---|---|---|---|
| Action space | continuous | continuous | continuous | discrete |
| Observation space | state | pixels | pixels | pixels |
| Exploration difficulty | low to moderate | high | moderate | moderate |
| Dynamics | deterministic / stochasitic | deterministic | deterministic | stochastic |

Figure 1: **Task domains included in RL Unplugged.** We include several open-source environments that are familiar to the community, as well as recent releases that push the limits of current algorithms. The task domains span key environment properties such as action space, observation space, exploration difficulty, and dynamics.

exploration in RL and the data requirements of online RL algorithms. Nevertheless, most real-world systems produce large amounts of data as part of their normal operation.

There is a resurgence of interest in offline methods for reinforcement learning,[1] that can learn new policies from logged data, without any further interactions with the environment due to its potential real-world impact. Offline RL can help (1) pretrain an RL agent using existing datasets, (2) empirically evaluate RL algorithms based on their ability to exploit a fixed dataset of interactions, and (3) bridge the gap between academic interest in RL and real-world applications.

Offline RL methods [e.g Agarwal et al., 2020, Fujimoto et al., 2018] have shown promising results on well-known benchmark domains. However, non-standardized evaluation protocols, differing datasets and lack of baselines make algorithmic comparisons difficult. Important properties of potential real-world application domains such as partial observability, high-dimensional sensory streams such as images, diverse action spaces, exploration problems, non-stationarity, and stochasticity are under-represented in the current offline RL literature. This makes it difficult to assess the practical applicability of offline RL algorithms.

The reproducibility crisis of RL [Henderson et al., 2018] is very evident in offline RL. Several works have highlighted these reproducibility challenges in their papers: Peng et al. [2019] discusses the difficulties of implementing the MPO algorithm, Fujimoto et al. [2019] mentions omitting results for SPIBB-DQN due to the complexity of implementation. On our part, we have had difficulty implementing SAC [Haarnoja et al., 2018]. We have also found it hard to scale BRAC [Wu et al., 2019] and BCQ [Fujimoto et al., 2018]. This does not indicate these algorithms do not work. Only that implementation details matter, comparing algorithms and ensuring their reproducibility is hard. The intention of this paper is to help in solving this problem by putting forward common benchmarks, datasets, evaluation protocols, and code.

The availability of large datasets with strong benchmarks has been the main factor for the success of machine learning in many domains. Examples of this include vision challenges, such as ImageNet [Deng et al., 2009] and COCO [Veit et al., 2016], and game challenges, where simulators produce hundreds of years of experience for online RL agents such as AlphaGo [Silver et al., 2016] and the OpenAI Five [Berner et al., 2019]. In contrast, lack of datasets with clear benchmarks hinders the similar progress in RL for real-world applications. This paper aims to correct this such as to facilitate collaborative research and measurable progress in the field.

To this end, we introduce a novel collection of task domains and associated datasets together with a clear evaluation protocol. We include widely-used domains such as the DM Control Suite [Tassa et al., 2018] and Atari 2600 games [Bellemare et al., 2013], but also domains that are still challenging for strong online RL algorithms such as real-world RL (RWRL) suite tasks [Dulac-Arnold et al., 2020] and DM Locomotion tasks [Heess et al., 2017, Merel et al., 2019a,b, 2020]. By standardizing the environments, datasets, and evaluation protocols, we hope to make research in offline RL more reproducible and accessible. We call our suite of benchmarks "RL Unplugged", because offline RL methods can use it without any actors interacting with the environment.

This paper offers four main contributions: (i) a unified API for datasets (ii) a varied set of environments (iii) clear evaluation protocols for offline RL research, and (iv) reference performance baselines. The datasets in RL Unplugged enable offline RL research on a variety of established online RL environments without having to deal with the exploration component of RL. In addition, we intend our evaluation protocols to make the benchmark more fair and robust to different hyperparameter choices compared to the traditional methods which rely on online policy selection. Moreover, releasing the datasets with a proper evaluation protocols and open-sourced code will also address the reproducibility issue in RL [Henderson et al., 2018]. We evaluate and analyze the results of several SOTA RL methods on each task domain in RL Unplugged. We also release our datasets in an easy-to-use unified API that makes the data access easy and efficient with popular machine learning frameworks.

## 2 RL Unplugged

The RL Unplugged suite is designed around the following considerations: to facilitate ease of use, we provide the datasets with a unified API which makes it easy for the practitioner to work with all data in the suite once a general pipeline has been established. We further provide a number of baselines including state-of-the art algorithms compatible with our API.[2]

### 2.1 Properties of RL Unplugged

Many real-world RL problems require algorithmic solutions that are general and can demonstrate robust performance on a diverse set of challenges. Our benchmark suite is designed to cover a range of properties to determine the difficulty of a learning problem and affect the solution strategy choice. In the initial release of RL Unplugged, we include a wide range of task domains, including Atari games and simulated robotics tasks. Despite the different nature of the environments used, we provide a unified API over the datasets. Each entry in any dataset consists of a tuple of state ($s_t$), action ($a_t$), reward ($r_t$), next state ($s_{t+1}$), and the next action ($a_{t+1}$). For sequence data, we also provide future states, actions, and rewards, which allows for training recurrent models for tasks requiring memory. We additionally store metadata such as episodic rewards and episode id. We chose the task domains to include tasks that vary along the following axes. In Figure 1, we give an overview of how each task domain maps to these axes.

**Action space**  We include tasks with both discrete and continuous action spaces, and of varying action dimension with up to 56 dimensions in the initial release of RL Unplugged.

**Observation space**  We include tasks that can be solved from the low-dimensional natural state space of the MDP (or hand-crafted features thereof), but also tasks where the observation space consists of high-dimensional images (*e.g.,* Atari 2600). We include tasks where the observation is recorded via an external camera (third-person view), as well as tasks in which the camera is controlled by the learning agent (e.g. robots with egocentric vision).

**Partial observability & need for memory**  We include tasks in which the feature vector is a complete representation of the state of the MDP, as well as tasks that require the agent to estimate the state by integrating information over horizons of different lengths.

**Difficulty of exploration**  We include tasks that vary in terms of exploration difficulty for reasons such as dimension of the action space, sparseness of the reward, or horizon of the learning problem.

**Real-world challenges**  To better reflect the difficulties encountered in real systems, we also include tasks from the Real-World RL Challenges [Dulac-Arnold et al., 2020], which include aspects such as action delays, stochastic transition dynamics, or non-stationarities.

The characteristics of the data is also an essential consideration, including the behavior policy used, data diversity, *i.e.,* state and action coverage, and dataset size. RL Unplugged introduces datasets that cover those different axes. For example, on Atari 2600, we use large datasets generated across training of an off-policy agent, over multiple seeds. The resulting dataset has data from a large mixture of policies. In contrast, we use datasets from fixed sub-optimal policies for the RWRL suite.

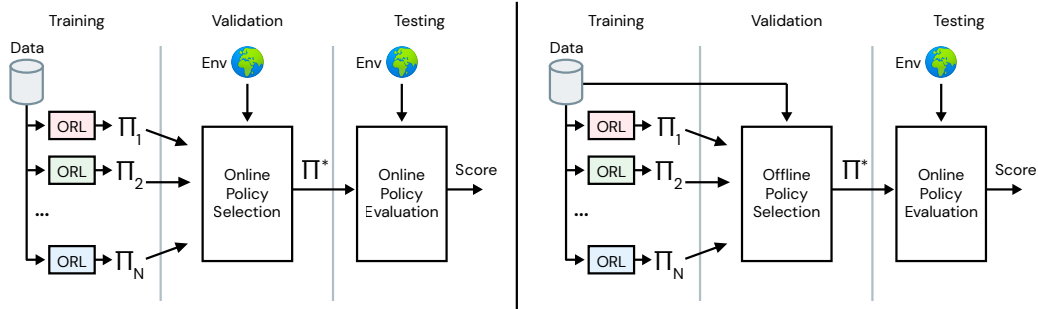

Figure 2: **Comparison of evaluation protocols.** (left) Evaluation using **online policy selection** allows us to isolate offline RL methods, but gives overly optimistic results because they allow perfect policy selection. (right) Evaluation using **offline policy selection** allows us to see how offline RL performs in situations where it is too costly to interact with the environment for validation purposes; a common scenario in the real-world. We intend our benchmark to be used for both.

## 2.2 Evaluation Protocols

In a strict offline setting, environment interactions are not allowed. This makes hyperparameter tuning, including determining when to stop a training procedure, difficult. This is because we cannot take policies obtained by different hyperparameters and run them in the environment to determine which ones receive higher reward (we call this procedure **online policy selection**).[3] Ideally, offline RL would evaluate policies obtained by different hyperparameters using only logged data, for example using offline policy evaluation (OPE) methods [Voloshin et al., 2019] (we call this procedure **offline policy selection**). However, it is unclear whether current OPE methods scale well to difficult problems. In RL Unplugged we would like to evaluate offline RL performance in both settings.

Evaluation by online policy selection (see Figure 2 (left)) is widespread in the RL literature, where researchers usually evaluate different hyperparameter configurations in an online manner by interacting with the environment, and then report results for the best hyperparameters. This enables us to evaluate offline RL methods in isolation, which is useful. It is indicative of performance given perfect offline policy selection, or in settings where we can validate via online interactions. This score is important, because as offline policy selection methods improve, performance will approach this limit. But it has downsides. As discussed before, it is infeasible in many real-world settings, and as a result it gives an overly optimistic view of how useful offline RL methods are today. Lastly, it favors methods with more hyperparameters over more robust ones.

Evaluation by offline policy selection (see Figure 2 (right)) has been less popular, but is important as it is indicative of robustness to imperfect policy selection, which more closely reflects the current state of offline RL for real-world problems. However it has downsides too, namely that there are many design choices including what data to use for offline policy selection, whether to use value functions trained via offline RL or OPE algorithms, which OPE algorithm to choose, and the meta question of how to tune OPE hyperparameters. Since this topic is still under-explored, we prefer not to specify any of these choices. Instead, we invite the community to innovate to find which offline policy selection method works best.

Importantly, our benchmark allows for evaluation in both online and offline policy selection settings. For each task, we clearly specify if it is intended for online vs offline policy selection. For offline policy selection tasks, we use a naive approach which we will describe in Section 4. We expect future work on offline policy selection methods to improve over this naive baseline. If a combination of offline RL method and offline policy selection can achieve perfect performance across all tasks, we believe this will mark an important milestone for offline methods in real-world applications.

Table 1. **DM Control Suite tasks.** We reserved five tasks for online policy selection (top) and the rest four are reserved for the offline policy selection (bottom). See Appendix E for reasoning behind choosing this particular task split.

| Environment | No. episodes | Act. dim. |
|---|---|---|
| Cartpole swingup | 40 | 1 |
| Cheetah run | 300 | 6 |
| Humanoid run | 3000 | 21 |
| Manipulator insert ball | 1500 | 5 |
| Walker stand | 200 | 6 |
| Finger turn hard | 500 | 2 |
| Fish swim | 200 | 5 |
| Manipulator insert peg | 1500 | 5 |
| Walker walk | 200 | 6 |

Table 2. **DM Locomotion tasks.** We reserved four tasks for online policy selection (top) and the rest three are reserved for the offline policy selection (bottom). See Appendix E for reasoning behind choosing this particular task split.

| Environment | No. episodes | Seq. length | Act. dim. |
|---|---|---|---|
| Humanoid corridor | 4000 | 2 | 56 |
| Humanoid walls | 4000 | 40 | 56 |
| Rodent gaps | 2000 | 2 | 38 |
| Rodent two tap | 2000 | 40 | 38 |
| Humanoid gaps | 4000 | 2 | 56 |
| Rodent bowl escape | 2000 | 40 | 38 |
| Rodent mazes | 2000 | 40 | 38 |

## 3 Tasks

For each task domain we give a description of the tasks included, indicate which tasks are intended for online vs offline policy selection, and provide a description of the corresponding data. Let us note that we have not modified how the rewards are computed in the original environments we used to generate the datasets. For the details of those reward functions, we refer to the papers where the environments were introduced first.

### 3.1 DM Control Suite

DeepMind Control Suite [Tassa et al., 2018] is a set of control tasks implemented in MuJoCo [Todorov et al., 2012]. We consider a subset of the tasks provided in the suite that cover a wide range of difficulties. For example, *Cartpole swingup* a simple task with a single degree of freedom is included. Difficult tasks are also included, such as *Humanoid run, Manipulator insert peg, Manipulator insert ball*. *Humanoid run* involves complex bodies with 21 degrees of freedom. And *Manipulator insert ball/peg* have not been shown to be solvable in any prior published work to the best of our knowledge. In all the considered tasks as observations we use the default feature representation of the system state, consisting of proprioceptive information such as joint positions and velocity, as well as additional sensors and target position where appropriate. The observation dimension ranges from 5 to 67.

**Data Description** Most of the datasets in this domain are generated using D4PG. For the environments *Manipulator insert ball* and *Manipulator insert peg* we use V-MPO [Song et al., 2020] to generate the data as D4PG is unable to solve these tasks. We always use 3 independent runs to ensure data diversity when generating data. All methods are run until the task is considered solved. For each method, data from the entire training run is recorded. As offline methods tend to require significantly less data, we reduce the sizes of the datasets via sub-sampling. In addition, we further reduce the number of successful episodes in each dataset by $2/3$ so as to ensure the datasets do not contain too many successful trajectories. See Table 1 for the size of each dataset. Each episode in this dataset contains 1000 time steps.

### 3.2 DM Locomotion

These tasks are made up of the corridor locomotion tasks involving the CMU Humanoid [Tassa et al., 2020], for which prior efforts have either used motion capture data [Merel et al., 2019a,b] or training from scratch [Song et al., 2020]. In addition, the DM Locomotion repository contains a set of tasks adapted to be suited to a virtual rodent [Merel et al., 2020]. We emphasize that the *DM Locomotion* tasks feature the combination of challenging high-DoF continuous control along with perception from rich egocentric observations.

**Data description** Note that for the purposes of data collection on the CMU humanoid tasks, we use expert policies trained according to Merel et al. [2019b], with only a single motor skill module from

Table 3: **Atari games.** We have 46 games in total in our Atari data release. We reserved 9 of the games for online policy selection (top) and the rest of the 37 games are reserved for the offline policy selection (bottom).

| | | | | |
|---|---|---|---|---|
| BEAMRIDER | DOUBLEDUNK | MS. PACMAN | ROAD RUNNER | ZAXXON |
| DEMONATTACK | ICE HOCKEY | POOYAN | ROBOTANK | |
| ALIEN | BREAKOUT | FROSTBITE | NAME THIS GAME | TIME PILOT |
| AMIDAR | CARNIVAL | GOPHER | PHOENIX | UP AND DOWN |
| ASSAULT | CENTIPEDE | GRAVITAR | PONG | VIDEO PINBALL |
| ASTERIX | CHOPPER COMMAND | HERO | Q*BERT | WIZARD OF WOR |
| ATLANTIS | CRAZY CLIMBER | JAMES BOND | RIVER RAID | YARS REVENGE |
| BANK HEIST | ENDURO | KANGAROO | SEAQUEST | |
| BATTLEZONE | FISHING DERBY | KRULL | SPACE INVADERS | |
| BOXING | FREEWAY | KUNG FU MASTER | STAR GUNNER | |

motion capture that is reused in each task. For the rodent task, we use the same training scheme as proposed by Merel et al. [2020]. For the CMU humanoid tasks, each dataset is generated by 3 online methods whereas each dataset of the rodent tasks is generated by 5 online methods. Similarly to the control suite, data from entire training runs is recorded to further diversify the datasets. Each dataset is then sub-sampled and the number of its successful episodes reduced by $2/3$. Since the sensing of the surroundings is done by egocentric cameras, all datasets in the locomotion domain include per-timestep egocentric camera observations of size $64 \times 64 \times 3$. The use of egocentric observation also renders some environments partially observable and therefore necessitates recurrent architectures. We therefore generate sequence datasets for tasks that require recurrent architectures. For dataset sizes and sequence lengths of see Table 2.

## 3.3 Atari 2600

The Arcade Learning environment (ALE) [Bellemare et al., 2013] is a suite consisting of a diverse set of 57 Atari 2600 games (Atari57). It is a popular benchmark to measure the progress of online RL methods, and Atari has recently also become a standard benchmark for offline RL methods [Agarwal et al., 2020, Fujimoto et al., 2019] as well. In this paper, we are releasing a large and diverse dataset of gameplay following the protocol described by Agarwal et al. [2020], and use it to evaluate several discrete RL algorithms.

**Data Description** The dataset is generated by running an online DQN agent and recording transitions from its replay during training with sticky actions [Machado et al., 2018]. As stated in [Agarwal et al., 2020], for each game we use data from five runs with 50 million transitions each. States in each transition include stacks of four frames to be able to do frame-stacking with our baselines.

In our release, we provide experiments on the 46 of the Atari games that are available in OpenAI gym. OpenAI gym implements more than 46 games, but we only include games where the online DQN's performance that has generated the dataset was significantly better than the random policy. We provide further information about the games we excluded in Appendix F. Among our 46 Atari games, we chose nine to allow for online policy selection. Specifically, we ordered all games according to the their difficulty,[4] and picked every fifth game as our offline policy section task to cover diverse set of games in terms of difficulty. In Table 3, we provide the full list of games that we decided to include in RL Unplugged.

## 3.4 Real-world Reinforcement Learning Suite

Dulac-Arnold et al. [2019, 2020] identify and evaluate respectively a set of 9 challenges that are bottlenecks to implementing RL algorithms, at scale, on applied systems. These include high-dimensional state and action spaces, large system delays, system constraints, multiple objectives, handling non-stationarity and partial observability. In addition, they have released a suite of tasks called `realworldrl-suite`[5] which enables a practitioner to verify the capabilities of their algorithm on domains that include some or all of these challenges. The suite also defines a set of standardized challenges with varying levels of difficulty. As part of the "RL Unplugged" collection, we have

generated datasets using the 'easy' combined challenges on four tasks: Cartpole Swingup, Walker Walk, Quadruped Walk and Humanoid Walk.

**Data Description**   The datasets were generated as described in Section 2.8 of [Dulac-Arnold et al., 2020]; note that this is the first data release based on those specifications. We used either the *no challenge* setting, which includes unperturbed versions of the tasks, or the *easy combined challenge* setting (see Section 2.9 of [Dulac-Arnold et al., 2020]), where data logs are generated from an environment that includes effects from combining all the challenges. Although the *no challenge* setting is identical to the control suite, the dataset generated for it is different as it is generated from fixed sub-optimal policies. These policies were obtained by training 3 seeds of distributional MPO [Abdolmaleki et al., 2018] until convergence with different random weight initializations, and then taking snapshots corresponding to roughly $75\%$ of the converged performance. For the *no challenge* setting, three datasets of different sizes were generated for each environment by combining the three snapshots, with the total dataset sizes (in numbers of episodes) provided in Table 4. The procedure was repeated for the *easy combined challenge* setting. Only the "large data" setting was used for the combined challenge to ensure the task is still solvable. We consider all RWRL tasks as online policy selection tasks.

Table 4: **real-world Reinforcement Learning Suite dataset sizes.** Size is measured in number of episodes, with each episode being 1000 steps long.

|                | Cartpole swingup | Walker walk | Quadruped walk | Humanoid walk |
|----------------|------------------|-------------|----------------|---------------|
| Small dataset  | 100              | 1000        | 100            | 4000          |
| Medium dataset | 200              | 2000        | 200            | 8000          |
| Large dataset  | 500              | 5000        | 500            | 20000         |

# 4   Baselines

We provide baseline results for a number of published algorithms for both continuous (DM Control Suite, DM Locomotion), and discrete action (Atari 2600) domains. We will open-source implementations of our baselines for the camera-ready. We follow the evaluation protocol presented in Section 2.2. Our baseline algorithms include behavior cloning (BC [Pomerleau, 1989]); online reinforcement learning algorithms (DQN [Mnih et al., 2015], D4PG [Barth-Maron et al., 2018], IQN [Dabney et al., 2018]); and recently proposed offline reinforcement learning algorithms (BCQ [Fujimoto et al., 2018], BRAC [Wu et al., 2019], RABM [Siegel et al., 2020], REM [Agarwal et al., 2020]). Some algorithms only work for discrete or continuous actions spaces, so we only evaluate algorithms in domains they are suited to. Detailed descriptions of the baselines and our implementations (including hyperparameters) are presented in Section A in the supplementary material.

**Naive approach for offline policy selection**   For the tasks we have marked for offline policy selection, we need a strategy that does not use online interaction to select hyperparameters. Our naive approach is to choose the set of hyperparameters that performs best overall on the online policy selection tasks from the same domain. We do this independently for each baseline. This approach is motivated by how hyperparameters are often chosen in practice, by using prior knowledge of what worked well in similar domains. If a baseline algorithm drops in performance between online and offline policy selection tasks, this indicates the algorithm is not robust to the choice of hyperparameters. This is also cheaper than tuning hyperparameters individually for all tasks, which is especially relevant for Atari. For a given domain, a baseline algorithm and a hyperparameter set, we compute the average[6] score over all tasks allowing online policy selection. The best hyperparameters are then applied to all offline policy selection tasks for this domain. The details of the experimental protocol and the final hyperparameters are provided in the supplementary material.

## 4.1   DM Control Suite

In Figure 4, we compare baselines across the online policy selection tasks (left) and offline policy selection tasks (right). A table of results is included in Section B of the supplementary material. For the simplest tasks, such as Cartpole swingup, Walker stand, and Walker walk, where the performance

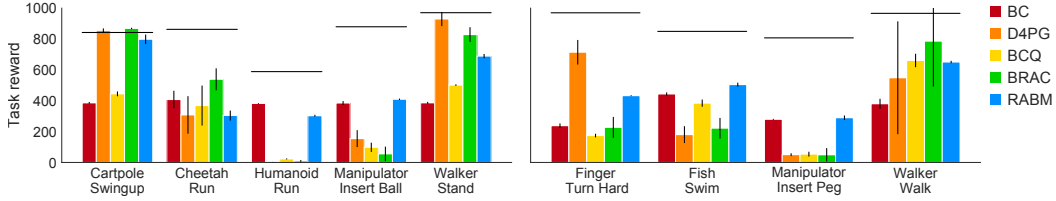

Figure 4: **Baselines on DM Control Suite.** (left) Performance using evaluation by online policy selection. (right) Performance using evaluation by offline policy selection. Horizontal lines for each task show 90th percentile of task reward in the dataset. Note that D4PG, BRAC, and RABM perform equally well on easier tasks e.g. Cartpole swingup. But BC, and RABM perform best on harder tasks e.g. Humanoid run.

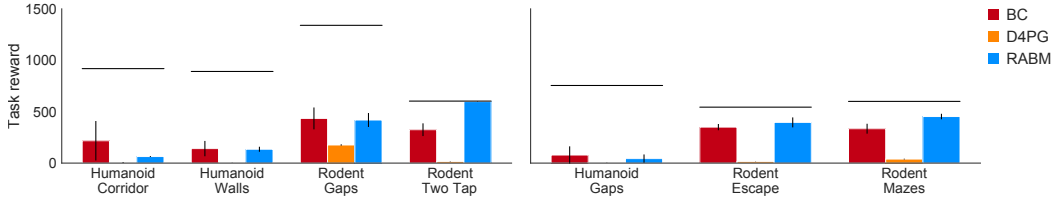

Figure 5: **Baselines on DM Locomotion.** (left) Performance using evaluation by online policy selection. (right) Performance using evaluation by offline policy selection. Horizontal lines for each task show 90th percentile of task reward in the dataset. The trend is similar to the harder tasks in DM Control Suite, i.e. BC and RABM perform well, while D4PG performs poorly.

of offline RL is close to that of online methods, D4PG, BRAC and RABM are all good choices. But the picture changes on the more difficult tasks, such as Humanoid run (which has high dimension action spaces), or Manipulator insert ball and manipulator insert peg (where exploration is hard). Strikingly, in these domains BC is actually among the best algorithms alongside RABM, although no algorithm reaches the performance of online methods. This highlights how including tasks with diverse difficulty conditions in a benchmark gives a more complete picture of offline RL algorithms.

## 4.2 DM Locomotion

In Figure 5, we compare baselines across the online policy selection tasks (left) and offline policy selection tasks (right). A table of results is included in Section C of the supplementary material. This task domain is made exclusively of tasks that are high action dimension, hard exploration, or both. As a result the stark trends seen above continue. BC, and RABM perform best, and D4PG performs quite poorly. We also could not make BCQ or BRAC perform well on these tasks, but we are not sure if this is because these algorithms perform poorly on these tasks, or if our implementations are missing a crucial detail. For this reason we do not include them. This highlights another key problem in online and offline RL. Papers do not include key baselines because the authors were not able to reproduce them, see eg [Peng et al., 2019, Fujimoto et al., 2019]. By releasing datasets, evaluation protocols and baselines, we are making it easier for researchers such as those working with BCQ to try their methods on these challenging benchmarks.

## 4.3 Atari 2600

In Figure 6, we present results for Atari using normalized scores. Due to the large number of tasks, we aggregate results using the median as done in [Agarwal et al., 2020, Hessel et al., 2018] (individual scores are presented in Appendix D). These results indicate that DQN is not very robust to the choice of hyperparameters. Unlike REM or IQN, DQN's performance dropped significantly on the offline policy selection tasks. BCQ, REM and IQN perform at least as well as the best policy in our training set according to our metrics. In contrast to other datasets (Section 4.1 and 4.2), BC performs poorly on this dataset. Surprisingly, the performance of off-the-shelf off-policy RL algorithms is competitive and even surpasses BCQ on offline policy selection tasks. Combining behavior regularization methods (*e.g.*, BCQ) with robust off-policy algorithms (REM, IQN) is a promising direction for future work.

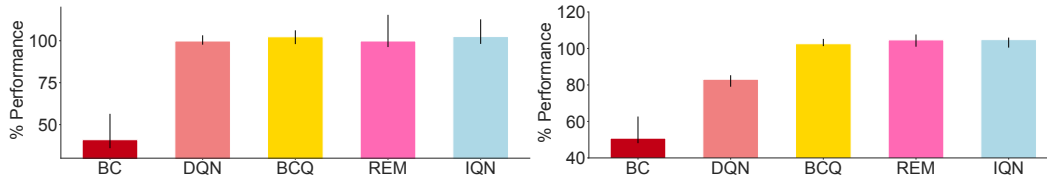

Figure 6: **Baselines on Atari.** (left) Performance using evaluation by online policy selection. (right) Performance using evaluation by offline policy selection. The bars indicate the median normalized score, and the error bars show a bootstrapped estimate of the $[25, 75]$ percentile interval for the median estimate computed across different games. The score normalization is done using the best performing policy among the mixture of policies that generated the offline Atari dataset (see Appendix H for details).

## 5   Related Work

There is a large body of work focused on developing novel offline reinforcement learning algorithms [Fujimoto et al., 2018, Wu et al., 2019, Agarwal et al., 2020, Siegel et al., 2020]. These works have often tested their methods on simple MDPs such as grid worlds [Laroche et al., 2017], or fully observed environments were the state of the world is given [Fujimoto et al., 2018, Wu et al., 2019, Fu et al., 2020]. There has also been extensive work applying offline reinforcement learning to difficult real-world domains such as robots [Cabi et al., 2019, Gu et al., 2017, Kalashnikov et al., 2018] or dialog [Henderson et al., 2008, Pietquin et al., 2011, Jaques et al., 2019], but it is often difficult to do thorough evaluations in these domains for the same reason offline RL is useful in them, namely that interaction with the environment is costly. Additionally, without consistent environments and datasets, it is impossible to clearly compare these different algorithmic approaches. We instead focus on a range of challenging simulated environments, and establishing them as a benchmark for offline RL algorithms. There are two works similar in that regard. The first is [Agarwal et al., 2020] which release DQN Replay dataset for Atari 2600 games, a challenging and well known RL benchmark. We have reached out to the authors to include this dataset as part of our benchmark. The second is [Fu et al., 2020] which released datasets for a range of control tasks, including the Control Suite, and dexterous manipulation tasks. Unlike our benchmark which includes tasks that test memory and representation learning, their tasks are all from fully observable MDPs, where the physical state information is explicitly provided.

## 6   Conclusion

We are releasing RL Unplugged, a suite of benchmarks covering a diverse set of environments, and datasets with an easy-to-use unified API. We present a clear evaluation protocol which we hope will encourage more research on offline policy selection. We empirically evaluate several state-of-art offline RL methods and analyze their results on our benchmark suite. The performance of the offline RL methods is already promising on some control suite tasks and Atari games. However, on partially-observable environments such as the locomotion suite the offline RL methods' performance is lower. We intend to extend our benchmark suite with new environments and datasets from the community to close the gap between real-world applications and reinforcement learning research.

### Acknowledgments and Disclosure of Funding

We appreciate the help we received from George Tucker and Ofir Nachum, firstly for sharing their BRAC implementation, and also running it on our DM Control Suite datasets which we reported as our baseline in this paper. We would like to thank Misha Denil for his insightful comments and feedback on our initial draft. We would like to thank Claudia Pope and Sarah Henderson for helping us in terms of arranging meetings and planning of the project.

### Broader Impact

Online methods require exploration by having a learning agent interact with an environment. In contrast, offline methods learn from fixed dataset of previously logged environment interactions.

This has three positive consequences: 1) Offline approaches are more straightforward in settings where allowing an agent to freely explore in the environment is not safe. 2) Reusing offline data is more environmentally friendly by reducing computational requirements, because in many settings exploration is the dominant computational cost and requires large-scale distributed RL algorithms. 3) Offline methods may be more accessible to the wider research community, insofar as researchers who do not have sufficient compute resources for online training from large quantities of simulated experience can reproduce results from research groups with more resources, and improve upon them.

But offline approaches also have potential drawbacks. Any algorithm that learns a policy from data to optimize a reward runs the risk of producing behaviors reflective of the training data or reward function. Offline RL is no exception. Current and future machine learning practitioners should be mindful of where and how they apply offline RL methods, with particular thought given to the scope of generalization they can expect of a policy trained on a fixed dataset.

## Footnotes

*Indicates joint first authors.

[1]Sometimes referred to as 'Batch RL,' but in this paper, we use 'Offline RL'.

[2]See our github project page for the details of our API (`https://github.com/deepmind/deepmind-research/tree/master/rl_unplugged`).

[3]Sometimes referred to as online model selection, but we choose policy selection to avoid confusion with models of the environment as used in model based RL algorithms.

[4] The details of how we decide the difficulty of Atari games are provided in Appendix G.

[5] See `https://github.com/google-research/realworldrl_suite` for details.

[6]We use the arithmetic mean with the exception of Atari where we use median following [Hessel et al., 2018].

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
