[Supplementary Material]

# Supplementary material

## A Detailed description of baselines

### A.1 Continuous Baselines

For DM Control Suite tasks (which only has feature observations) we used an MLP with 8 layers of size 1024 using residual connections and instance normalization after every two layers to encode features.

For Locomotion tasks (which also include pixel observations), the image inputs were first preprocessed by a Resnet and the embedding was concatanating to the features observations, before feeding it into the MLP descibed above.

For the tasks of sequential nature (Rodent Two Tap, Rodent Escape and Rodent Mazes), the MLP was followed by two LSTMs with hidden size 1024 each.

The output of the MLP is then fed into a linear layer, which predicts parameters of the Mixture of 5 Multivariate Gaussian distributions, which is used as the final policy output. The mixture distribution is used here to capture the multimodal nature of data in some of the environments (e.g. in Locomotion Humanoid experiments, the data consists of very diverse way of running).

When training the policy we let the variances of every Gaussian is adjusted, but when evaluating the policy, we fix the variance to be $1e-4$, since we found that reducing the noise can greatly improve the performance.

In Table 5, we show the hyperparameters shared among our baselines. Next, we describe each baseline separately and provide individual hyperparameters. We used the nine online policy selection environments (see Tables 1 and 2) to choose these hyperparameters.

Table 5: **Continuous control experiments Hyperparameters.** The top section of the table corresponds to the hyperparameters shared across agents (the ResNet hyperparameters are only applicable to the Locomotion experiments), while the bottom section of the table correspond to the hyperparameters which differ across agents.

| Hyperparameter | setting (shared across agents) |
|---|---|
| Discount factor | 0.99 |
| Target network update period | every 100 updates |
| resnet: num blocks | 2, 2, 2 |
| resnet: channels | 16, 32, 32 |
| resnet: filter size | $3 \times 3$ |
| resnet: stride | 2 |

| Hyperparameter | D4PG | ABM & BC |
|---|---|---|
| resnet: hidden units | 512 | 64 |
| resnet: activation function | ReLu | Instance norm + Elu |
| $Q$-network distributional parameters: range | $[-150, 150]$ | $[0, 100]$ |
| $Q$-network distributional parameters: num atoms | 51 | 21 |

**BC** Behavior Cloning [Pomerleau, 1989] is a supervised learning algorithm in which the policy learns to mimic the behavior policy by learning a mapping between observations and actions, without consideration of reward. We use the Adam optimizer [Kingma and Ba, 2015] with the learning rate of $1e-4$. We used batch size 128 when using recurrent networks and batch size 1024 when not.

**D4PG** Distributed Distributional Deep Deterministic Policy Gradient [Barth-Maron et al., 2018] is an online RL algorithm repurposed for offline RL. D4PG's distributional critic estimates the distribution of discounted cumulative returns of the current policy, and its policy learns to take actions with high values under the critic. For both the actor and critic, we use the Adam optimizer [Kingma and Ba, 2015] with the learning rate of $1e-4$. We use D4PG implemented in Acme Hoffman et al. [2020], following their network architectures and hyper-parameters. We used batch size 256 for the experiments.

**BCQ**   Batch-Constrained deep Q-learning [Fujimoto et al., 2018]. In addition to the critic and policy, BCQ trains a generative model trained to mimic the behavior policy that generated the dataset. Continuous BCQ trains a variational autoencoder and uses that VAE to decide the actions to take in the target network. We use the exact same network architecture and the algorithm that is described in [Fujimoto et al., 2018]. We used batch size 1024 for the experiments.

**BRAC**   Behavior Regularized Actor Critic [Wu et al., 2019] is an actor critic algorithm where the actor is encouraged to stay close to the behavior policy. BRAC estimates the KL divergence between the policy and the behavior policy; the policy is penalized for large divergence via what the authors call value penalty. We use the exact same network architecture as described in the original paper. We use the Adam optimizer [Kingma and Ba, 2015] with critic learning rate is set to $1e-3$. We use behavioral cloning (trained for 300000 learner steps and with learning rate $5e-4$) to estimate the behavioral distribution which is used to compute the KL-divergence. Batch size is set to 256 for all BRAC experiments. We swept over the policy learning rate parameter (on the grid $[1e-5, 1e-4, 3e-5]$) as well as the KL penalty parameter $\alpha$ (on the grid $[0.1, 0.3, 1.0]$).

**RABM**   Distributional Advantage-weighted Behavior Model is a slight modification of ABM [Siegel et al., 2020] which uses advantage weighted regression to learn a prior policy to which the policy is constrained to stay close to via MPO [Abdolmaleki et al., 2018]. RABM additionally introduces distributional critics as well as recurrence capabilities; the latter for solving partially observable environments. The policy is also trained to take actions that achieve high critic values. We also chose to use different network architectures to follow those used for BC. We use most of the original hyper-parameters but modified learning rates. For training the prior, policy and critic, we use Adam optimizers [Kingma and Ba, 2015] with the learning rate of $1e-4$.

Both for the prior policy and the final policy we use the same architecture as for the BC policy described above, except that in the last layer we use Multivariate Gaussian distribution instead of a mixture of such distributions, since the MPO-like part of the ABM update rule is specifically designed for Gaussian policies. For the critic, we use the same architecture (ResNet for processing visual inputs, residual MLP for processing features concatenated with image embeddings, LSTMs on top for environment requiring recurrence), but concatenate actions with the features and in the last layer output logits of the discrete distribution that defines the distributional $Q$-function.

### A.2   Discrete Baselines

In our Atari experiments we have used the same network architecture that was proposed in [Mnih et al., 2013]. For all our discrete baselines, we have performed a hyperparameter search for the learning rate from the grid $[5e-5, 1e-4, 5e-5]$ and used the Adam optimizer [Kingma and Ba, 2015] with the default $\beta$ and $\epsilon$ hyperparameters in Tensorflow 2.

In Table 6, we show the hyperparameters shared among our baselines. Next, we describe each baseline separately and provide individual hyperparameters and respective grid search values. We used nine games to evaluate the agents with online policy selection and the rest of the games we could only evaluate the agent with offline policy selection as described before.

**BC**   Behavior Cloning [Pomerleau, 1989]. See description above in the continuous baselines section. We used learning rate of $0.00003$ for the evaluation by offline policy selection. When evaluating BC, we use the max action from the policy head.

**DQN**   We used the standard Deep Q-Networks [Mnih et al., 2015] off-policy learning algorithm developed for online RL as a baseline for offline RL. Our results as well as previously reported results [Agarwal et al., 2020] have proven that DQN trained with Adam optimizer is a strong offline RL baseline. We found that learning rate of $0.00003$ was performing the best for offline DQN by evaluation with online policy selection. We also used double DQN [Van Hasselt et al., 2016] in our Q-learning loss which we found it to be useful in our preliminary experiments.

**IQN**   Implicit Quantile Networks [Dabney et al., 2018] is an online distributional RL algorithm that approximates the return distribution using the full quantile function, a continuous map from probabilities to returns. We used this baseline since it is a SOTA distributional method on Atari, and Agarwal et al. [2020] has previously shown that distributional methods can perform competitively in

Table 6: **Atari experiments Hyperparameters.** The top section of the table corresponds to the shared hyperparameters of the offline RL methods and the bottom section of the table contrasts the hyperparameters of Online vs Offline DQN.

| Hyperparameter | setting (for both variations) |
|---|---|
| Discount factor | 0.99 |
| Mini-batch size | 256 |
| Target network update period | every 2500 updates |
| Evaluation $\epsilon$ | $0.4^8$ |
| $Q$-network: channels | 32, 64, 64 |
| $Q$-network: filter size | $8 \times 8, 4 \times 4, 3 \times 3$ |
| $Q$-network: stride | 4, 2, 1 |
| $Q$-network: hidden units | 512 |
| Training Steps | 2M learning steps |
| Hardware | Tesla V100 GPU |
| Replay Scheme | Uniform |

| Hyperparameter | Online | Offline |
|---|---|---|
| Min replay size for sampling | 20,000 | - |
| Training $\epsilon$ (for $\epsilon$-greedy exploration) | 0.01 | - |
| $\epsilon$-decay schedule | 250K steps | - |
| Fixed Replay Memory | No | Yes |
| Replay Memory size | 1M steps | 2M steps |
| Double DQN | No | Yes |

the offline RL setting. We found learning rate of $1e - 4$ to work best with IQN when evaluating with online policy selection. We have evaluated 8, 16 and 32 $\tau$ samples by online policy selection. We found that 16 $\tau$ samples performs the best and we have shown the performance of IQN by online policy selection with respect to different $\tau$ values is discussed further in Section I.

**BCQ** Batch-Constrained deep Q-learning [Fujimoto et al., 2019]. The discrete variant of BCQ is very similar to the continuous variant. Discrete BCQ uses that generative model trained in a supervised manner as a constraint to decide which actions to take in the target network. The discrete BCQ has a threshold hyper-parameter to determine when to trust the action taken by the generative model. We have done a grid search to find the best threshold hyperparameter. The grid we used for the threshold is, $[0.25, 0.5, 0.75, 1.0]$. According to our our online policy selection, BCQ with learning rate of $0.0001$ and threshold of $0.5$ performed the best. In Section J, we have discussed and shown the sensitivity of BCQ with respect to the threshold hyperparameter.

**REM** Random Ensemble Mixture [Agarwal et al., 2020] uses multiple parameterized Q-functions to estimate the Q-values. The key intuition behind REM is that if one has access to multiple estimates of Q-values, then a random convex combination of the Q-value estimates is also an estimate for Q-values. Accordingly, in each training step, REM randomly combines multiple Q-value estimates and uses this random combination for robust training. We have used a random ensemble of 4 DQN networks in our implementation. According to our our online policy selection, we found the learning rate of $1e - 4$ to be performing the best, and used that on our offline policy selection games as well.

## B  DM Control Suite results

Detailed results for DM Control Suite are presented in Table 7.

Table 7: **DM Control suite results.**

| Task name | Baselines | | | | |
|---|---|---|---|---|---|
| | **BC** | **D4PG** | **BCQ** | **BRAC** | **RABM** |
| Cartpole swingup | 386±6 | **856±13** | 445±16 | **869±4** | 798±31 |
| Cheetah run | 408±57 | 308±122 | 369±130 | **539±71** | 304±32 |
| Humanoid run | **382±3** | 1.72±1.66 | 22.8±3.5 | 9.62±5.75 | 303±6 |
| Manipulator insert ball | 385±13 | 154±55 | 98.0±29.8 | 55.6±46.8 | **409±5** |
| Walker stand | 386±7 | **930±46** | 502±5 | 829±48 | 689±14 |
| Finger turn hard | 238±15 | **714±80** | 174±12 | 227±68 | 433±3 |
| Fish swim | 444±10 | 180±55 | 384±23 | 222±67 | **504±13** |
| Manipulator insert peg | **279±3** | 50.4±9.2 | 54.0±15.6 | 49.5±43.2 | **290±15** |
| Walker walk | 380±32 | **549±366** | **661±44** | **786±294** | **651±8** |

## C  DM Locomotion results

Detailed results for DM Locomotion Suite are presented in Table 8.

Table 8: **DM Locomotion results.**

| Task name | Baselines | | |
|---|---|---|---|
| | **BC** | **D4PG** | **RABM** |
| Humanoid corridor | **220±194** | 4.39±4.15 | **64.5±3.8** |
| Humanoid walls | **139±77** | 2.71±1.05 | **132±25** |
| Rodent gaps | **464±137** | 176±7 | **421±71** |
| Rodent two tap | 326±60 | 16.6±2.6 | **599±3** |
| Humanoid gaps | 35.9±50.6 | 2.36±1.26 | **80.0±8.6** |
| Rodent bowl escape | 389±3 | 16.2±1.1 | **440±5** |
| Rodent mazes | 344±48 | 40.2±3.9 | **476±2** |

## D  Atari 2600 Results

Detailed unnormalized results for Atari 2600 Suite are presented in Table 9.

In Figure 7 normalized results for each individual game and baseline are presented.

Table 9: **Atari 2600 results.** Unnormalized evaluation scores. For each difficulty level on Atari, we first report the results on the games where we evaluated the agent with online policy selection, and then the ones on which we only evaluated the agents with offline policy selection.

| Name | BC | DQN | IQN | BCQ | REM |
|---|---|---|---|---|---|
| BeamRider | 1.48K ± 0.34K | 1.81K ± 0.18K | **3.02K ± 0.87K** | 1.99K ± 0.02K | 2.20K ± 0.29K |
| DemonAttack | 7.6K ± 3.0K | 11.0K ± 3.1K | 15.5K ± 8.4K | **19.3K ± 7.4K** | 17.0K ± 7.6K |
| DoubleDunk | -16.4 ± 2.5 | **-17.9 ± 5.1** | **-16.7 ± 3.9** | **-12.9 ± 5.3** | -17.9 ± 4.3 |
| IceHockey | -5.63 ± 1.99 | **-2.88 ± 2.93** | -4.65 ± 2.03 | -2.51 ± 1.02 | **-1.16 ± 1.04** |
| MsPacman | **4.04K ± 0.93K** | 2.47K ± 0.27K | **4.39K ± 0.58K** | 3.08K ± 0.54K | 3.15K ± 0.48K |
| Pooyan | 3.85K ± 0.21K | 3.18K ± 1.03K | **5.00K ± 0.63K** | 4.20K ± 0.42K | **4.47K ± 0.68K** |
| RoadRunner | 19.0K ± 12.4K | **31.7K ± 26.9K** | 44.7K ± 12.3K | **57.4K ± 0.8K** | 56.5K ± 1.7K |
| Robotank | 15.7 ± 8.0 | **55.7 ± 10.8** | 42.7 ± 17.1 | **60.7 ± 2.2** | **60.5 ± 3.3** |
| Zaxxon | 0.01K ± 0.01K | 6.05K ± 1.58K | 0.87K ± 0.91K | **9.43K ± 1.47K** | **8.30K ± 1.18K** |
| Alien | **2.67K ± 1.03K** | 1.69K ± 0.26K | **2.86K ± 0.44K** | 2.09K ± 0.33K | 1.73K ± 0.25K |
| Amidar | **256 ± 122** | 224 ± 28 | **351 ± 173** | 254 ± 43 | 214 ± 31 |
| Asterix | 2.96K ± 1.02K | 1.52K ± 0.13K | **5.71K ± 0.23K** | 1.93K ± 0.20K | 4.89K ± 0.31K |
| Assault | 1.81K ± 0.13K | 1.94K ± 0.24K | 2.18K ± 0.15K | 2.26K ± 0.29K | **3.07K ± 0.91K** |
| Atlantis | 2.39M ± 0.88M | **3.02M ± 0.52M** | 2.71M ± 0.88M | **3.20M ± 0.24M** | **3.36M ± 0.19M** |
| BattleZone | 4.8K ± 2.6K | **25.6K ± 4.7K** | 16.5K ± 3.7K | **25.4K ± 2.5K** | **26.2K ± 3.6K** |
| BankHeist | **1.05K ± 0.09K** | 0.05K ± 0.07K | **1.11K ± 0.06K** | 0.27K ± 0.10K | 0.16K ± 0.04K |
| Boxing | 83.9 ± 4.0 | 96.3 ± 0.4 | 95.8 ± 0.9 | **97.2 ± 0.4** | **97.3 ± 0.4** |
| Breakout | 235 ± 16 | 324 ± 26 | 314 ± 9 | **375 ± 12** | 362 ± 15 |
| Carnival | **3.92K ± 1.73K** | 1.45K ± 0.54K | **4.82K ± 0.21K** | 4.31K ± 0.35K | 2.08K ± 0.66K |
| Centipede | 1.07K ± 0.33K | 1.25K ± 0.18K | **1.83K ± 0.30K** | 1.43K ± 0.20K | 0.81K ± 0.10K |
| ChopperCommand | 0.66K ± 0.17K | 2.25K ± 0.32K | 0.83K ± 0.13K | **3.95K ± 1.24K** | **3.61K ± 0.50K** |
| CrazyClimber | 123M ± 1M | 23M ± 15M | **126M ± 2M** | 28M ± 15M | 42M ± 2M |
| Enduro | 0.72K ± 0.27K | 1.21K ± 0.27K | 1.70K ± 0.16K | 1.39K ± 0.25K | **3.65K ± 0.87K** |
| FishingDerby | -7.4 ± 20.3 | 17.0 ± 3.1 | 20.8 ± 3.1 | **28.9 ± 0.9** | **29.3 ± 2.4** |
| Freeway | **21.8 ± 14.7** | 15.4 ± 3.6 | **24.7 ± 13.8** | 16.9 ± 2.9 | 7.2 ± 5.4 |
| Frostbite | 0.78K ± 0.55K | **3.23K ± 0.42K** | 2.63K ± 0.52K | **3.52K ± 0.44K** | 3.07K ± 0.27K |
| Gopher | 4.9K ± 1.9K | 2.4K ± 1.0K | **11.3K ± 1.0K** | **8.7K ± 4.6K** | 3.7K ± 0.2K |
| Gravitar | 20 ± 16 | 500 ± 64 | 235 ± 91 | **580 ± 40** | **424 ± 196** |
| Hero | 13.9K ± 0.2K | 5.2K ± 3.0K | **16.2K ± 2.9K** | 13.2K ± 4.9K | **14.0K ± 4.6K** |
| Jamesbond | 237 ± 42 | 490 ± 164 | **699 ± 272** | 438 ± 191 | 369 ± 236 |
| Kangaroo | **5.69K ± 4.76K** | 0.82K ± 0.14K | **9.12K ± 2.14K** | 1.30K ± 0.53K | 1.21K ± 0.54K |
| KungFuMaster | 5.1K ± 5.6K | 16.1K ± 2.7K | **19.5K ± 3.7K** | 16.9K ± 1.1K | **19.4K ± 2.7K** |
| Krull | **8.50K ± 0.16K** | 7.48K ± 0.19K | **8.47K ± 0.27K** | 7.78K ± 0.60K | 7.98K ± 0.38K |
| NameThisGame | 4.1K ± 0.4K | 11.5K ± 0.2K | 9.9K ± 0.9K | 12.6K ± 0.3K | **13.0K ± 0.5K** |
| Phoenix | 2.94K ± 0.93K | **6.41K ± 2.91K** | 4.94K ± 0.35K | **6.62K ± 2.65K** | **7.48K ± 2.91K** |
| Pong | **18.9 ± 0.6** | 12.9 ± 4.2 | **19.2 ± 0.9** | 16.5 ± 2.8 | 16.5 ± 3.5 |
| Qbert | **12.6K ± 1.0K** | 10.6K ± 2.2K | **13.4K ± 0.9K** | 12.6K ± 1.4K | **13.1K ± 0.7K** |
| Riverraid | 6.0K ± 1.6K | 9.1K ± 2.4K | **13.0K ± 1.8K** | **14.2K ± 1.1K** | **14.2K ± 2.0K** |
| Seaquest | 0.15K ± 0.06K | 2.87K ± 1.71K | 1.67K ± 0.53K | 5.41K ± 1.58K | **5.91K ± 2.39K** |
| SpaceInvaders | 0.79K ± 0.31K | 2.71K ± 0.08K | **2.84K ± 0.12K** | **2.92K ± 0.07K** | 2.81K ± 0.08K |
| StarGunner | 3.0K ± 2.3K | 1.6K ± 0.9K | **39.4K ± 5.4K** | 2.5K ± 0.2K | 7.5K ± 1.6K |
| TimePilot | 1.95K ± 0.98K | **5.31K ± 0.50K** | 3.14K ± 0.96K | **5.18K ± 0.41K** | 4.49K ± 0.42K |
| UpNDown | 16.3K ± 3.4K | 14.6K ± 5.6K | **32.3K ± 22.3K** | **32.5K ± 22.5K** | 27.6K ± 7.9K |
| VideoPinball | 27M ± 19M | 82M ± 61M | 102M ± 85M | 103M ± 74M | **313M ± 111M** |
| WizardOfWor | 0.73K ± 0.58K | 2.30K ± 0.51K | 1.40K ± 0.83K | **4.68K ± 1.43K** | 2.73K ± 0.88K |
| YarsRevenge | 19.1K ± 6.6K | 24.9K ± 2.5K | **28.4K ± 2.9K** | **29.1K ± 1.1K** | 23.1K ± 2.9K |

Figure 7: **Atari normalized performance.** Atari Normalized Performance improvement (in %), per game, of (from top to bottom) offline BC, offline DQN, offline IQN, offline BCQ, and offline REM trained for 2 million learner steps. The normalized online score for each game is 0.0 and 1.0 for the random agent and the fully-trained online DQN, respectively. These results show that the best performing offline agents (IQN, REM) are able to improve upon the fully-trained DQN on approximately half of the games only.

## E   On the Choice of DM Control Suite and DM Locomotion Splits

When working on offline RL algorithms in a strict setting (when using environment interactions is not allowed even for choosing hyperparameters of the algorithm), researchers typically use offline policy evaluation (OPE) methods to tune hyperparameters. However, OPE methods themselves might require tuning, so it is desirable to have both online tasks – tasks for which we can evaluate a policy in the environment and thus to tune the hyperparameters of the OPE method, and offline tasks – tasks which can be used to evaluate the final performance. However, if the two sets of tasks are completely distinct, it might be hard to transfer knowledge and hyperparameters from the online set of tasks to the offline set. This is why when splitting tasks in DM Control Suite (see Table 1) and DM Locomotion (see Table 2) into online policy selection and offline policy selection subsets, we tried to make the environments in the two sets similar, e.g. Walker stand (an environment form online policy selection subset) is similar to Walker walk (an environment form offline policy selection subset). And while Catrpole swingup is not similar to Finger turn hard in nature, at least they have similar action specs.

## F   Atari Data Selection Details

We have excluded games from our suite such as AirRaid, JourneyEscape since they are not in atari 57. We didn't include Elevator Action, Berzerk, JourneyEscape, MontezumaRevenge, PitFall, Private Eye, Skiing, Solaris and Venture since they are very hard exploration games. We omitted Asteroids, Bowling, Tutankham and Tennis since our online DQN generating the data performed very poorly on these games.

## G   Atari Game Difficulty Categorization

We categorized Atari games in difficulty based on the performance comparison of offline DQN and the best policy from online DQN. In Figure 8, we show the performance of offline-DQN, which is run over all the Atari games described by [Agarwal et al., 2020], with the average behavior and best policy in the dataset. If the performance of offline DQN is consistently below behavior policy, we categorize the task as hard. If the performance of the agent is in between the best policy and the average behavior policy in the dataset, we consider the task as medium difficulty. If the offline DQN agent's performance is in between the performance of average behavior policy and if the offline DQN can perform better than the best policy in the dataset, we consider that game easy.

## H   Atari Results Normalization and Environment Details

Following Agarwal et al. [2020], we report the absolute normalized % performance with respect to normalized $= 100 \times \frac{\text{score} - \text{random\_score}}{\text{best\_score} - \text{random\_score}}$ where "normalized" corresponds to the absolute normalized percentage performance of the method with respect to the best snapshot of online DQN "best_score" that is used to generate the dataset. "score" corresponds to the episodic reward that the offline agent receives in the environment, "random_score" corresponds to the score of an random agent for the games in a single number.

We follow Agarwal et al. [2020] to set environment details like sticky action probability (see Table 10).

Table 10: **Atari 2600 Environment Details.**

| Name | Value |
| --- | --- |
| Sticky action probability | 0.25 |
| Grey-scaling | True |
| Observation down-sampling | (84, 84) |
| Frames stacked | 4 |
| Frame skip (Action repetitions) | 4 |
| Reward clipping | [-1, 1] |
| Terminal condition | Game Over |
| Max frames per episode | 108K |

Figure 8: **Atari games difficulty categorization.** We categorized the Atari games in terms of difficulty based on the learning curves of the offline DQN trained over all games. The dashed line for each game indicates the performance of the average policy and the straight line shows the performance of the best policy in the dataset.

# I Atari IQN Ablation Study

In Figure 9, we show the sensitivity of the IQN to the number of $\tau$ samples with online policy evaluation. In our experiments, we found out that $16$ $\tau$ samples achieves the best performance on the online policy selection games where for online IQN agent $8\tau$ samples performed better.

Figure 9: **IQN $\tau$ samples sensitivity**. The sensitivity of the IQN to the number of $\tau$ samples with online policy evaluation. The square dark markers for each threshold denotes the median % Performance of the best online DQN, and the error bars show a bootstrapped estimate of the $[25, 75]$ percentile interval for the mean estimate. In our experiments, we found out that $16$ $\tau$ samples achieves the best performance on the online policy selection games.

# J Atari BCQ Ablation Study

The discrete BCQ has a threshold hyper-parameter to determine when to trust the action taken by the generative model. In Figure 10, we have shown the sensitivity of BCQ to that hyperparameter on the Atari dataset. Overall, we found that the threshold of $0.5$ works the best with online policy selection, please note that by setting threshold to $0$, one would recover the offline DQN (which performs worse than BCQ).

Figure 10: **BCQ threshold sensitivity.** We show the sensitivity of BCQ to the threshold hyperparameter. The square dark markers for each threshold denotes the median % Performance of the best online DQN, and the error bars show a bootstrapped estimate of the $[25, 75]$ percentile interval for the mean estimate.