[Reviews · NeurIPS 2020]

Review 1

Summary and Contributions: This paper introduces a collection of task domains, datasets, and benchmark results for offline reinforcement learning. It presents an evaluation of how different common offline RL methods perform on the proposed datasets.

Strengths: As mentioned by the authors, there is certainly a need for better offline (batch) reinforcement learning benchmarks due to the difficulty of reproducing paper results and understanding whether a proposed algorithm is truly better than another. In this regard, the proposed benchmark is a welcome addition - datasets for offline RL are uncommon while new algorithms that claim state-of-the-art are released and publicized at increasing frequencies. There are a large number of tasks and domains provided (with the exception of robotic manipulation, which would be a welcome addition) and the empirical evaluation is thorough. Release of the datasets and the code to reproduce all results would be an important step in the right direction for the field of offline RL. Some of the authors' findings are also surprising - such as Behavioral Cloning being a strong baseline on more challenging offline RL settings.

Weaknesses: My biggest concern is that most of the datasets seem homogenous in terms of data collection sources. Most seem to consist of experience collected from a handful of RL algorithm runs. In real world settings, data collection could take place from heterogenous sources of data, such as humans. In that regard, it seems prudent to keep the task domains fixed and provide datasets that vary the quality of dataset sources along different dimensions (e.g. random policy, degree of suboptimality of the experiences, the diversity and multimodality in the solution strategies and exploration strategies used by policies that collected the data). Data collection through humans could also be considered, as done in prior works like this one (https://arxiv.org/abs/1811.02790) or this one (https://arxiv.org/abs/1909.12200). It is important for offline RL benchmarks to provide and compare against data generated through more natural and realistic data collection sources. Indeed, the D4RL benchmark also provides datasets with multiple data collection strategies for a fixed task (https://arxiv.org/abs/2004.07219). While offline policy selection certainly has merit (especially in real world scenarios), the logic behind selecting the tasks for offline policy selection per set of datasets seems arbitrary (except for Atari - the authors explain the rationale there). Furthermore, it would be useful to benchmark the set of "offline policy selection" tasks using online policy selection as well to compare the two (for example, as a comparison for the right side of Figures 4 and 5) - this is part of the value of having simulated benchmarks and domains. Reproducibility is one of the key features that are highlighted as the motivation for this dataset and benchmark, but there are some potential issues here. From the supplementary material, it appears that certain network architectures are rather large and could be demanding in terms of compute resources necessary to reproduce experiments - it would be useful for the authors to comment on such requirements. It seems rather excessive (in terms of number of parameters) for the BC baseline to use an 8-layer MLP with 1024 hidden size per layer, and the use of residual connections and instance norm is rather unconventional compared to other works that include BC as a baseline. The choice of a 5-mode GMM policy parametrization is also unconventional, as is the rather large batch sizes (1024 for BC with flat policies). I also noticed that BRAC uses the same network architectures as the paper, which might put it at an unfair disadvantage compared to the BC baseline. There are also several mentions to a unified API for the datasets and it is even listed as a contribution, but the authors do not mention additional details about this API. Post-rebuttal: my biggest concern was ease of reproducing the results - the authors have addressed most of my other concerns. I would still also prefer that ablations be conducted for some unconventional choices (such as the larger batch size used, and the use of GMM policy instead of a unimodal Gaussian), and I think it is also important that code be released to reproduce the results on the benchmark, as well as hardware details needed for reproduction. I will maintain my score, and recommendation for acceptance.

Correctness: Overall, the experiments are sound and mostly well motivated, apart from the weaknesses mentioned above.

Clarity: No issues with clarity - the paper is well-written. Minor comment - line 190 - "but we only games" seems like a grammar mistake / typo.

Relation to Prior Work: There are a few missing citations - some large-scale offline datasets do already exist for robotic manipulation domains see this paper (https://arxiv.org/abs/1811.02790) and this one (https://arxiv.org/abs/1911.04052). Furthermore, IRIS (https://arxiv.org/abs/1911.05321) is an offline RL algorithm for robotic manipulation.

Reproducibility: Yes

Additional Feedback: Additional insight for how and why BC and RABM outperform other methods on harder tasks would be great. It is especially interesting because if BC performs well, then the reward function is not helping improve task performance. In section 4.2, I really appreciate the honesty on not getting a subset of algorithms to work well on certain datasets, and I agree that this highlights the reproducibility crisis, but both BCQ and BRAC provide public implementations - did those also perform poorly on the benchmark tasks? For the RWRL suite, the utility of the "no challenge" dataset is still unclear. If it is indeed just the data collection source that is different, then perhaps it should be organized differently (see my comment on providing multiple datasets per task).


Review 2

Summary and Contributions: In this paper a a suite of benchmarks for offline reinforcement learning is introduced. The importance of offline policy selection is stressed. Several existing RL benchmarks are included, the results of several algorithms are presented. The authors intend to extend the benchmark suite with new environments and datasets to close the gap between real-world applications and reinforcement learning research.

Strengths: It is a good idea to develop a benchmark suite for offline (also called batch) reinforcement learning. It is important to stress the importance of offline policy selection. The objective of increasing the applicability of reinforcement learning for real world applications, especially in an industrial environment, is also very important.

Weaknesses: So far no benchmark for offline (or batch) reinforcement learning has been included that has the ambition to have the characteristics of a real application. After all, the tasks: Cartpole Swingup, Walker Walk, Quadruped Walk and Humanoid Walk, have very little in common with real applications. AFTER FEEDBACK To the answer of the authors to my objections; "R3: On data from real applications. We only focus on sim datasets because it is easier to evaluate the resulting policies using publicly available environments. However, working with real-world data requires OPE advances or other ways of scoring the results. We are investigating this possibility and would like to include results from real-world applications in the future. We believe our existing tasks will be useful for developing offline RL methods for real-world applications. Notably, RWRL was designed to include real-world problems such as stuck sensors, delays, perturbations, etc." I would like to respond: I agree with the authors that simulations are better suited for benchmarks, especially when it comes to active learning or reinforcement learning, than real data or real environments. But the one does not exclude the other. You can use simulations that come very close to real application. Without these, a publication seems to me premature. My recommendation is therefore: please search for freely available simulations of real applications, there are such, especially in the industrial environment.

Correctness: Yes.

Clarity: Yes

Relation to Prior Work: When discussing off policy evaluation, a reference to A. Hans, et al.; „Agent self-assessment: Determining policy quality without execution.”, ADPRL 2011: 84-90 or A. Hans, “Advancing the Applicability of Reinforcement Learning to Autonomous Control”, PhD thesis, TU Illmenau, Germany, 2014 would be useful, as this is, to my knowledge, the first publication on this issue. It also explains in detail and in the same spirit as this paper the importance for real applications.

Reproducibility: Yes

Additional Feedback: The title of subsection 3.4 is "real-world Reinforcement Learning Suite". Is there a reason to use lower case for "real-world"? Please correct missing capital letters in the bibliography.


Review 3

Summary and Contributions: The authors present a suite of data sets for offline evaluation of reinforcement learning algorithms. I really like this idea and feel that this could be a timely and relevant contribution. The main contribution is that the authors propose a suite of relatively standardized tasks that is heterogeneous in terms of the data and the problem setting.

Strengths: - A pragmatic approach to a complex problem - A large number of tasks that can be extended in a crowdsourced fashion - A good selection of heterogeneous problems - A standardized evaluation protocol

Weaknesses: - the authors could explain the API (if any) for their framework better - maybe a comparison of models evaluated with such a static RL task suite and models evaluated with other standard RL frameworks would be nice

Correctness: does not apply here

Clarity: yes

Relation to Prior Work: yes, the related work section captures this well enough.

Reproducibility: Yes

Additional Feedback:


Review 4

Summary and Contributions: This paper contributes a benchmark for offline RL. The benchmark consists of a set of tasks as well as datasets of trajectories from each task. The metric for the benchmark is the on-policy performance of models trained solely on the off-policy data. The tasks vary along several axes including the type of observations and actions and stochasticity of dynamics. The benchmark does not include goal-conditioned environments.

Strengths: Strengths: - Offline RL is an important challenge is making RL useful for real world settings, where safety is often of vital concern. - Evaluation is done on both offline policy selection and online policy selection which encourages future research to work on the offline policy selection process. - The benchmark offers a variety of tasks. - The novelty of the benchmark lies in standardization the source of data for offline RL, as well as in releasing code for the policy selection phases.

Weaknesses: Weaknesses: - The descriptions of the tasks don’t all include how rewards are given/calculated. - Without running both offline and online policy selection on the same tasks, the opportunity to measure that gap in performance was lost. This seems like an obvious experiment to run. - Goal-conditioned tasks are not included in the benchmark.

Correctness: Yes

Clarity: The paper is easy to read and clear.

Relation to Prior Work: To my knowledge the relation to prior work is clear.

Reproducibility: Yes

Additional Feedback: Questions: - I would be interested in further analysis about the how the policy that generated the data to learn from affects performance, and whether certain off-policy algorithms are better at using data from some data-generated mechanisms over others. - Why were the tasks partitioned into on-line policy selection vs offline-policy selection? This aspect of the benchmark is independent of the task, each tasks could be run with both. --------------- After reading the other reviews and author response, I continue to find the paper of interest to the neurips community as long as the authors make the changes stated in the response.

[Author Response · NeurIPS 2020]

We would like to thank our reviewers for their thoughtful comments and feedback. We will add the missing citations and fix the typos pointed out by the reviewers.

**R1 and R4:** *On lack of API description in text.* Our open-sourced repository describes the API in detail, along with examples of agents using it. However, to preserve anonymity, we can not share the link to the repository. Additionally, we will present the API details in the supplementary material of the camera-ready version of our paper.

**R1 and R5:** *On adding results for online policy selection for the offline policy selection task.* We are concerned that if researchers perform online policy selection and offline policy selection on the same tasks, they may have privileged information and could accidentally bias offline policy selection algorithms towards good hyperparameters. So, we provided distinct tasks for online and offline policy selection in hopes of preventing this. But this choice may be overly cautious, and we are planning to run these experiments since it was suggested by both reviewers.

**R1:** *On homogeneous data sources.* We agree, offline RL datasets from heterogeneous sources are important, and we would like to include them. The main limitation was a practical one: for challenging tasks training RL agents to solve them, and collecting a large number of human demonstrations are both significant endeavors, and we only focused on one. Our most challenging tasks are locomotion tasks, which are not well suited for human demonstrations. Nevertheless, we believe that the datasets we currently provide are already useful to the community because: 1) they highlight differences between current algorithms, and 2) SOTA algorithms perform quite poorly on the harder tasks. In our tasks, we believe that the main challenge comes from the difficulty of the tasks due to aforementioned properties in our paper. But we believe this is an important direction for research as well.

**R1:** *On task selection, and defining splits.* Our rationale for choosing this particular task split was to ensure that online and offline tasks both cover easy and hard tasks equally (similar to Atari). We will add this rationale to the paper.

**R1:** *On architecture used for the BC and other baselines.* We highlight that common control benchmarks in the literature use smaller networks, because they have compact state representations. We use relatively large networks due to the complexity of the DM locomotion tasks. We adopted the same network architecture for consistency, but we agree smaller networks would be sufficient for the control suite tasks. Additionally, we chose to use the batch size of 1024 instead of smaller batch sizes because it runs faster on the hardware we used. Note however that we tried using smaller batch size (256) on a few environments and got identical results (though it took more wallclock time).

**R1:** *On unconventional GMM parametrization.* Offline RL algorithms should deal with datasets obtained from multiple policies as it would be a common case for real-life applications. Hence, our dm_control and locomotion datasets are generated by multiple agents that often behave quite differently from each other. As a result, we chose to use GMMs since they could better deal with this multimodality in the action space.

**R1:** *On using BCQ and BRAC public implementations.* We used the public implementations of BRAC from github and its hyperparameters tuned for the dm-control suite tasks. We made sure that our implementation of BCQ reproduces the results in the BCQ paper with the same architecture and hyperparameters.

**R1:** *On the utility of the "no challenge" dataset.* These datasets are generated from the same tasks as the perturbed RWRL environment, whereas for the control suite we use different tasks. Grouping the "no challenge" dataset together with the combined challenge data for RWRL allows us to examine the effect of the various RWRL challenges on the learning capability of offline RL methods.

**R3:** *On data from real applications.* We only focus on sim datasets because it is easier to evaluate the resulting policies using publicly available environments. However, working with real-world data requires OPE advances or other ways of scoring the results. We are investigating this possibility and would like to include results from real-world applications in the future. We believe our existing tasks will be useful for developing offline RL methods for real-world applications. Notably, RWRL was designed to include real-world problems such as stuck sensors, delays, perturbations, etc.

**R5:** *On the lack of details on how rewards are calculated for each tasks.* Our datasets use the original rewards associated with environments that were used to generate the data, and they are described in the documentation of the environments. We will also add these details to the supplementary material of our paper for completeness.

**R5:** *Goal-conditioned tasks are not included in the benchmark.* Thanks for the suggestion, we are planning to add goal-conditioned tasks in the future, in addition to dmlab and hard-eight datasets. We are open to contributions from the community, and hope that the RL community will contribute new tasks that RL Unplugged is lacking.

**R5:** *On the effect of data distribution.* We have an analysis of this on Atari and will include it in the supplementary material. We compared different models with respect to the changes in the stochasticity of the transition dynamics, state-action coverage, and reward distributions in the dataset. We found out that behavior regularized models are more robust to those changes.

[Meta-Review · NeurIPS 2020]

This paper presents a benchmark suite for offline RL, along with baseline results on this suite. There is a clear need for such a benchmark suite and all reviewers agree that this submission is a good first step towards fulfilling this need. Only R3 considers that it is not ready yet for acceptance, due to lacking more realistic benchmarks (i.e. closer to complex real world industrial applications). Although I agree that this is a valid concern (and I encourage the authors to follow R3’s advice and look up potential simulators that may fill up this gap), I also agree with other reviewers that the current state of the proposed benchmark should already be very useful to the research community. As a result, I recommend to accept this paper.